# Allele-specific gene expression can underlie altered transcript abundance in zebrafish mutants

Richard J White[1], Eirinn Mackay[2], Stephen W Wilson[2], Elisabeth M Busch-Nentwich[1,3]*

[1]Cambridge Institute of Therapeutic Immunology & Infectious Disease (CITIID), Department of Medicine, University of Cambridge, Cambridge, United Kingdom; [2]Department of Cell and Developmental Biology, University College London, London, United Kingdom; [3]School of Biological and Behavioural Sciences, Faculty of Science and Engineering, Queen Mary University of London, London, United Kingdom

*For correspondence:
e.busch-nentwich@qmul.ac.uk

Competing interest: The authors declare that no competing interests exist.

**Abstract** In model organisms, RNA-sequencing (RNA-seq) is frequently used to assess the effect of genetic mutations on cellular and developmental processes. Typically, animals heterozygous for a mutation are crossed to produce offspring with different genotypes. Resultant embryos are grouped by genotype to compare homozygous mutant embryos to heterozygous and wild-type siblings. Genes that are differentially expressed between the groups are assumed to reveal insights into the pathways affected by the mutation. Here we show that in zebrafish, differentially expressed genes are often over-represented on the same chromosome as the mutation due to different levels of expression of alleles from different genetic backgrounds. Using an incross of haplotype-resolved wild-type fish, we found evidence of widespread allele-specific expression, which appears as differential expression when comparing embryos homozygous for a region of the genome to their siblings. When analysing mutant transcriptomes, this means that the differential expression of genes on the same chromosome as a mutation of interest may not be caused by that mutation. Typically, the genomic location of a differentially expressed gene is not considered when interpreting its importance with respect to the phenotype. This could lead to pathways being erroneously implicated or overlooked due to the noise of spurious differentially expressed genes on the same chromosome as the mutation. These observations have implications for the interpretation of RNA-seq experiments involving outbred animals and non-inbred model organisms.

## Editor's evaluation

Zebrafish strains are typically considerably polymorphic. White and colleagues tested the hypothesis that genes in linkage with a mutant allele might show allele-specific expression differences and thus potentially confound the interpretation of mutant effects. Using a variety of mutant and wild-type alleles with sophisticated analysis of RNA-seq data in zebrafish embryos they demonstrate over-representation of expression changes of genes in linkage with the mutant allele on the same chromosome. The authors provide Gene Ontology analyses to demonstrate how the allele-specific expression differences may impact on the interpretation of differential gene expression caused by a mutation. The data are extensive, carefully analysed and of sufficient depth and quality to support their main claim of frequent occurrence of allele-specific gene expression in outcross experiments. The findings of this study will be of interest to geneticists working with zebrafish strains or with strains of other polymorphic species.

## Introduction

Large-scale genetic screens to identify gene function by randomly introducing mutations have been a staple of zebrafish genetics for several decades (*Driever et al., 1996*; *Haffter et al., 1996*; *Kettleborough et al., 2013*). The advent of RNA-sequencing (RNA-seq) has enabled investigators to estimate the location of such mutations in the genome, while also providing information regarding gene expression levels and affected cellular pathways in the mutants. The bioinformatics pipelines which process RNA-seq data to generate gene expression information focus on transcript abundance, differential splicing, and gene set enrichments, and, in general, genomic location is not considered when assessing genes that are differentially expressed (DE) in a mutant context. Here, we report that physical location can impact a gene's likelihood of being DE in mutant zebrafish.

In the typical protocol for introducing random point mutations, male zebrafish from a laboratory wild-type strain are treated with *N*-ethyl-*N*-nitrosourea (ENU) to mutagenise sperm (*Kettleborough et al., 2011*; *Mullins et al., 1994*). The mutagenised fish (G0) are mated with wild-type females to produce F1 offspring, each heterozygous at random novel mutation sites. F1 fish are outcrossed with wild types to produce clutches of F2 offspring, which are subsequently incrossed to produce F3 embryos. The F3 clutches contain the novel mutations in Mendelian ratios, and in a forward genetics approach are screened for recessive phenotypes of interest which appear in approximately 25% of embryos (*Mullins et al., 1994*). These embryos are referred to as 'mutants' whereas those without phenotypes are 'siblings'.

Mutant embryos are homozygous for a novel allele (the 'causative mutation') and due to genetic linkage, they are likely to be homozygous for alleles physically nearby on the chromosome. The location encompassing the causative mutation therefore lies in a region which is highly homozygous in mutants, yet heterozygous in siblings. This is referred to as linkage disequilibrium (LD). The region of high LD can be mapped using high-throughput sequencing and bioinformatics pipelines (*Mackay and Schulte-Merker, 2014*; *Minevich et al., 2012*; *Obholzer et al., 2012*) whereas prior efforts involved painstaking genotyping of simple sequence length polymorphisms and genome walks using bacterial or P1 artificial chromosome libraries or subsequently, microarrays (*Stickney et al., 2002*; *Zhang et al., 1998*).

All mapping processes rely on identification of polymorphic loci throughout the genome. Laboratory zebrafish strains have a high degree of intra-strain polymorphism (*Guryev et al., 2006*), but mapping is aided by the introduction of alleles from other strains. Thus, mutagenised males are often paired with females from a different strain. As a result, in a mapping cross, alleles in the mutants and siblings are inherited from two different strains. This remains true throughout the multiple generations that a mutant line is maintained in a laboratory.

In this study, we report that the highly polymorphic nature of zebrafish strains can lead to gene expression differences between mutant and sibling embryos through allele-specific expression (ASE). The effect of ASE is well documented across many species, and can be tissue- and condition-specific (*Ayroles et al., 2009*; *Doss et al., 2005*; *Fu et al., 2009*; *Battle et al., 2017*; *Huang et al., 2015*; *Kim-Hellmuth et al., 2020*; *Storey et al., 2005*). Here, this phenomenon manifests as a cluster of DE genes located near to the causative mutation site in many different unrelated mutant lines. The differential transcript levels of these local genes are likely due to expression differences between wild-type strains rather than altered transcription due to the mutation. We confirm the high prevalence of ASE in zebrafish in the SAT line which is derived from only two haplotypes. This observation has implications for researchers attempting to use differential expression to explain phenotypes of interest, not only in zebrafish, but also in other outbred model organisms, as these local genes may simply be a red herring.

## Results

### Differentially expressed genes are often enriched on the mutant chromosome

To map the causal mutations for a number of different mutants from forward genetic screens, we used RNA-seq and LD mapping, based on Cloudmap (*Minevich et al., 2012*). A representative LD mapping plot (taken from the mutant line *u426*) is shown in *Figure 1*. We observed a high degree of LD on chromosome 7 at approximately 22 Mbp, suggesting the phenotype-causing mutation is near this

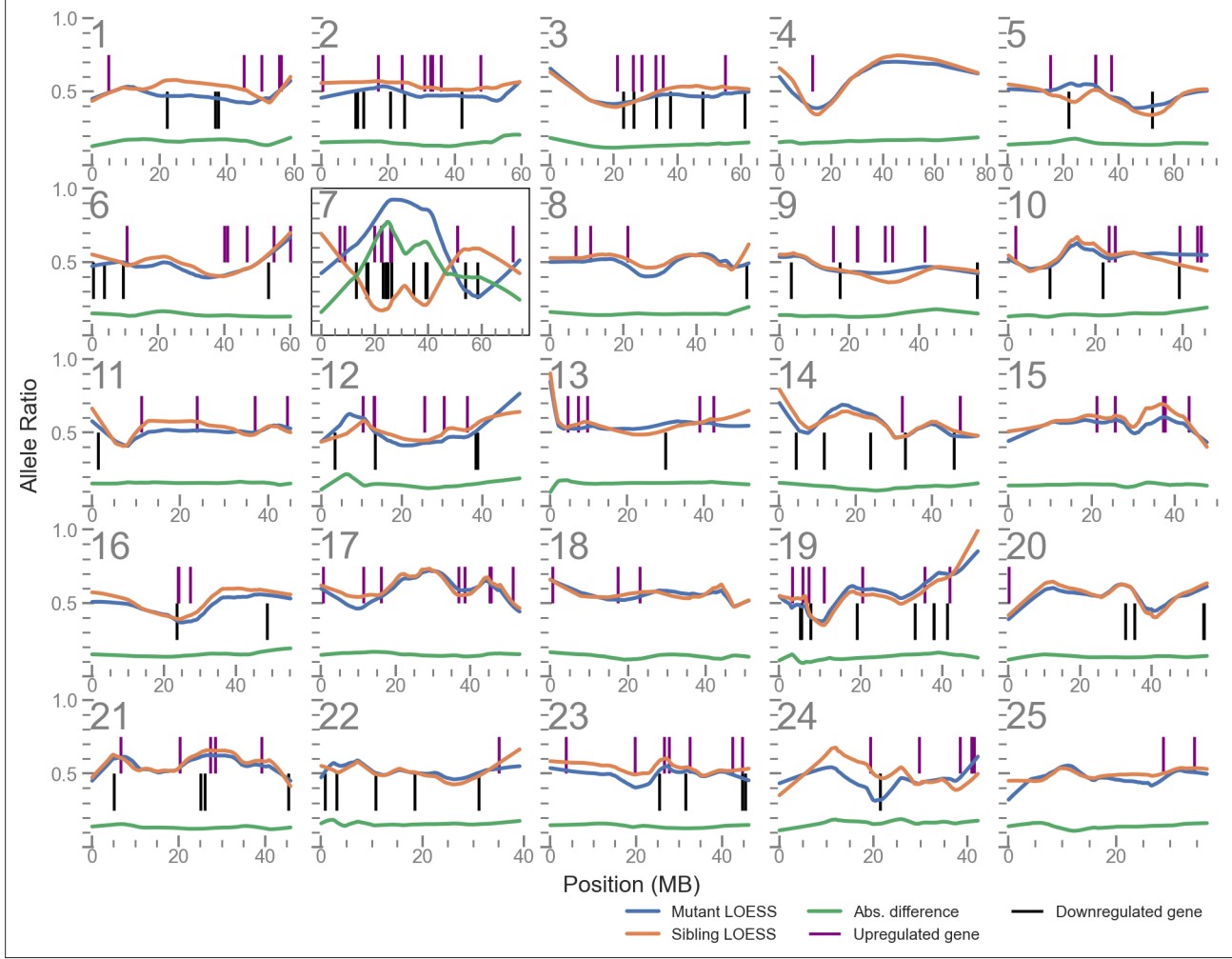

**Figure 1.** Linkage disequilibrium (LD) mapping plot of up- and downregulated genes in *u426* mutants shows a cluster of such genes local to the mutation site on chromosome 7. The plots for each of the 25 chromosomes shows the allele balance (proportion of reads containing the alternative allele) of each single nucleotide polymorphism (SNP) locus along with its physical position. The blue and orange lines are LOESS-smoothed averages of the data. The green line is the absolute difference of the mutant and sibling samples and is used to identify the region of highest LD. Vertical lines indicate the position of differentially expressed genes.

position. DESeq2 reported 209 genes as DE (adjusted p-value < 0.05) between mutants and siblings. Annotating the LD mapping plot with the position of these genes showed a cluster of DE genes near the LD mapping peak on chromosome 7. Indeed, we found 15 DE genes in an arbitrarily sized 20 Mbp window centred on the mapping peak at 22 Mbp, representing 7% of all DE genes. For comparison, a 20 Mbp window randomly sampled (1000 iterations) from the zebrafish genome contains approximately 1.4% of known genes.

We then used a logistic regression model to examine the effect of LD on the probability of an individual gene being DE. A summary of each line and the regression results are presented in *Table 1*. Of nine mutant lines analysed (*Supplementary file 1*), seven samples showed a significant, positive effect of LD (Benjamini/Hochberg adjusted p-value < 0.05). To help visualise the effect of LD on DE probability, we calculated an odds ratio for each sample by comparing the DE probability at the site of maximum LD with the probability at a site of median LD. In the most extreme case (the sample *nl14*), the likelihood of finding a DE gene near to the mutation site was over 100-fold higher than the likelihood of finding one at a random other location in the genome.

In parallel, we were analysing a separate catalogue of 3' tag sequencing experiments of zebrafish mutant lines (115 experiments), most of which were generated and made available as part of the Zebrafish Mutation Project (*Collins et al., 2015*; *Dooley et al., 2019*; *Kettleborough et al., 2013*).

**Table 1.** Summary of logistic regression results for RNA-sequencing (RNA-seq) analysed mutant lines.

Causative mutation shows the gene and location of the mutation site in lines where this has been confirmed empirically, otherwise the location is estimated from linkage disequilibrium (LD) data. Significance column indicates adjusted p-value (***: < 0.001, **: < 0.01; *: < 0.05). Odds ratio compares DE likelihood at maximum LD versus site of median LD. The nearby genes column shows the number of DE genes lying within a 20 Mbp window centred on the mutation site, and the percentage of these genes out of the total DE genes. In-table citations: [1](***Barlow et al., 2020***), [2](***Miesfeld et al., 2015***), [3](***Armant et al., 2016***). *nl14* line kindly provided by Alex Nechiporuk.

| Allele | Causative mutation | DE genes/total | Coefficient ± SEM | Sig. | Odds ratio | Nearby genes (%) |
|---|---|---|---|---|---|---|
| *nl14* | *lama1* unpublished (chr24, 41.6Mbp) | 12/31,664 | 9.09 ± 1.56 | *** | 118.5 | 3 (25%) |
| *la015577*[1] | *dmist* (chr5, 19.9 Mbp) | 157/31,199 | 6.84 ± 0.46 | *** | 55.8 | 23 (15%) |
| *u505*[1] | *dmist* (chr5, 19.9 Mbp) | 71/31,199 | 8.72 ± 0.72 | *** | 44.0 | 13 (18%) |
| *u757* | Unpublished (chr23, 22 Mbp) | 33/31,199 | 6.31 ± 2.13 | ** | 7.8 | 1 (3%) |
| *u534* | Not known (chr1, ~25 Mbp) | 87/31,664 | 4.83 ± 1.05 | *** | 5.4 | 4 (5%) |
| *u426* | Not known (chr7, ~22 Mbp) | 209/31,664 | 2.67 ± 0.48 | *** | 5.3 | 15 (7%) |
| *nl13*[2] | *yap1* (chr18, 37.2 Mbp) | 140/31,199 | 2.58 ± 1.57 | – | 2.3 | 4 (3%) |
| *sb55*[3] | *ache* (chr 7, 26.0 Mbp) | 348/24,558 | 3.77 ± 1.67 | * | 2.0 | 14 (4%) |
| *u535* | Not known (chr13, ~25 Mbp) | 294/31,663 | 0.35 ± 1.04 | – | 1.1 | 4 (1%) |

These were analysed for differential expression, producing a large collection of DE gene lists. We noticed that, often, the mutant chromosome had a large proportion of the total number of DE genes in the experiment. For example, comparing *mitfa*[w2/w2] embryos to siblings produces 116 DE genes, 48 of which are present on chromosome 6, which is the chromosome where *mitfa* is located (***Figure 2A***).

To investigate this, we tested for chromosomes that had an enrichment of DE genes under the null hypothesis that they are randomly distributed across the genome. In all, 60 chromosomes from 37 lines had a statistically significant enrichment of the DE genes (binomial test, Bonferroni adjusted $p < 0.05$). Of these, 44 were on the chromosome carrying the mutation being investigated in the experiment (***Supplementary file 2***). Of the other 16, 7 had an enrichment on chromosome 9. This was driven by expression of γ-crystallin genes (***Supplementary file 3***), which are expressed in the lens and present in a cluster on chromosome 9 (***Greiling et al., 2009***) that we have previously observed as being co-regulated (***White et al., 2017***). This suggests that the eyes are affected in some of the analysed mutants. Whether there was an enrichment of DE genes on the mutant chromosome did not depend on the total number of DE genes found in the experiment, although experiments with very high numbers of DE genes tended not to show an enrichment (***Figure 2B***).

In one experiment, we noticed that the differential expression of some genes was linked to one of the wild-type chromosomes in the experiment. This experiment was an intercross of two different *sox10* alleles, *t3* (***Dutton et al., 2001***) and *baz1* (***Carney et al., 2006***) sampled at 24 hr post-fertilisation (hpf). Embryos were genotyped for both *sox10* alleles, which allowed us to also track the wild-type chromosomes in the cross. We noticed that two of the genotypes had expression levels for some genes on the same chromosome as *sox10* that were different from the other two genotypes (***Figure 2C***). The groups with higher expression shared the wild-type chromosome from the *baz1/+* parent (***Figure 2C***, yellow chromosome) whereas the others shared the chromosome carrying the *baz1* allele (***Figure 2C***, blue chromosome). One explanation for this is that there is higher expression from the *si:ch73–308m11.1* allele on the wild-type chromosome (***Figure 2C***, yellow chromosome), which led us to hypothesise that the enrichment of DE genes on the mutant chromosome is not necessarily dependent on the mutant gene.

Our hypothesis is that ASE, that is, polymorphism-driven variation in expression levels of genes, is common across the genome. This would manifest as differential expression when a genomic locus

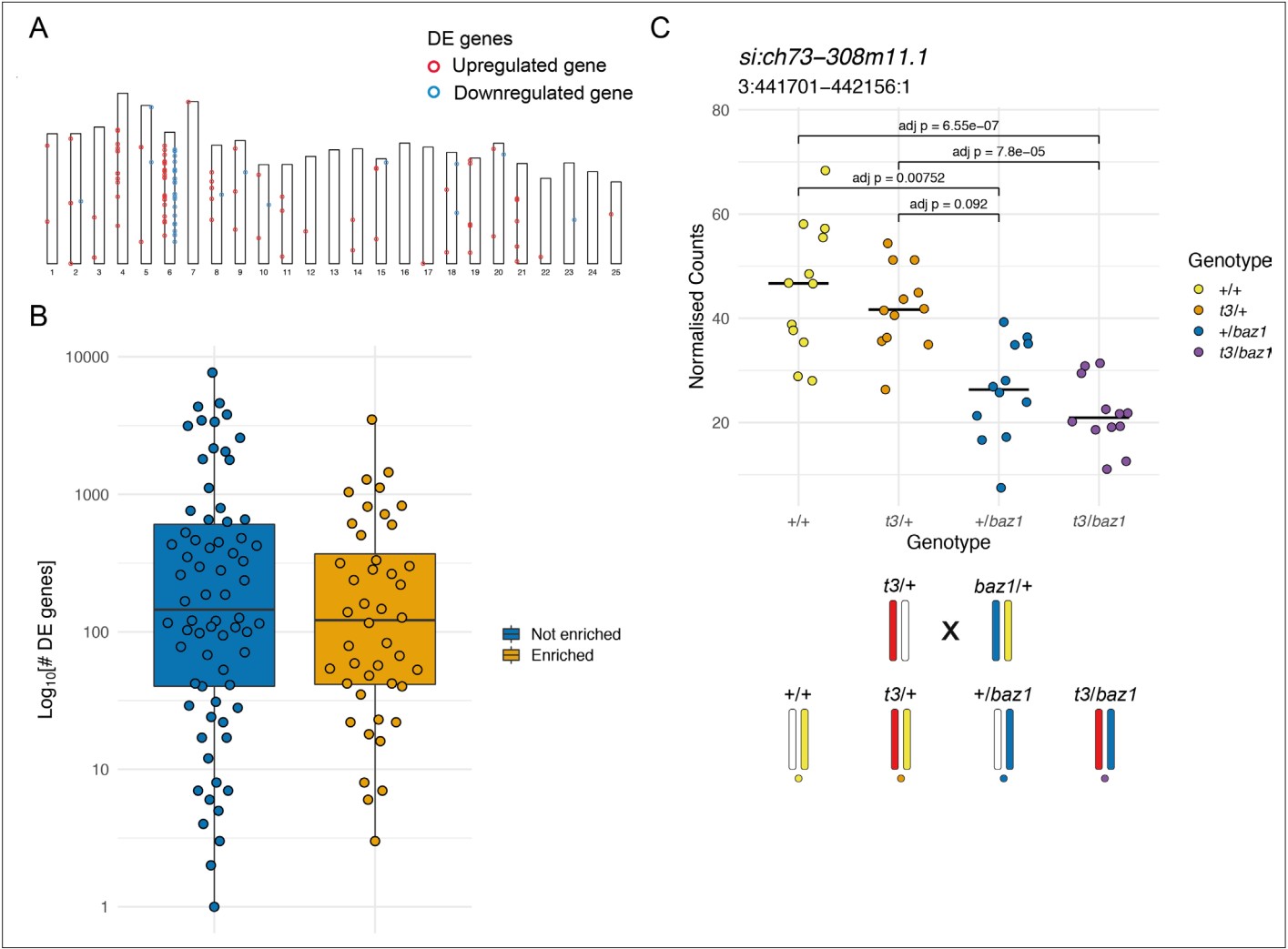

**Figure 2.** Enrichment of differentially expressed (DE) genes on the mutant chromosome. (**A**) Ideogram showing the locations of the DE genes in a *mitfa^w2* incross. Circles represent DE genes and are coloured red if the gene is upregulated in the mutant embryos and blue if it is downregulated. (**B**) Distribution of the total number of DE genes in experiments according to whether there is an enrichment on the mutant chromosome (orange) or not (blue), plotted on a log₁₀ scale. (**C**) Plot of normalised counts according to genotype in an intercross of two different *sox10* alleles. Yellow = wild type (+/+), orange = *sox10* t3 heterozygotes (t3/+), blue = *sox10 baz1* heterozygotes (+/baz1), purple = *sox10* t3, *baz1* compound heterozygotes (t3/baz1). The schematic below the plot shows the chromosomes contributing to each genotype. Embryos that share the wild-type allele inherited from the *baz1/+* parent (yellow chromosome) show higher expression levels.

The online version of this article includes the following source data for figure 2:

**Source data 1.** Genomic positions of differentially expressed (DE) genes.

**Source data 2.** DeTCT differentially expressed (DE) genes data.

**Source data 3.** Counts for si:ch73-308m11.1.

is driven to homozygosity in some individuals and the expression levels of genes in this locus are compared to those in individuals that are heterozygous, or homozygous for the other allele.

## ASE is common in a wild-type cross

To test the hypothesis that the over-representation of DE genes on the mutant chromosome can be driven by ASE independently of the mutated gene, we investigated gene expression in wild-type fish with defined haplotypes to enable easy identification of the different alleles in the cross. We used the SAT line, which was generated from an intercross of one fully sequenced double haploid AB fish and one fully sequenced double haploid Tübingen fish (*Howe et al., 2013*). This means that for any

position in the genome there are up to two possible alleles. The original haplotypes have recombined through the generations that the SAT line has been maintained by incrossing.

We incrossed two SAT fish, fin-clipped them to isolate DNA for exome sequencing, collected 96 morphologically normal embryos at 5 days post-fertilisation (dpf), extracted RNA from the individual embryos, and did RNA-seq on the 96 samples. We used the exome sequence of the SAT parent fish for this cross to call SNPs and identify regions that are either homozygous for the AB haplotype, homozygous for the Tübingen haplotype, or heterozygous. Using the RNA-seq reads and SNPs identified in the parental exome data, we genotyped the embryos at locations that distinguish the AB and Tübingen haplotypes. Aggregating these data in 1 Mbp regions allowed us to determine the haplotypes of each individual embryo. We identified regions of the parental genomes where at least two genotypes, and thus potentially ASE, are possible in the offspring (informative regions) and where we had sufficient read depth to unambiguously identify the haplotypes in the offspring. We grouped the 96 RNA-seq samples according to their haplotype in that region (*Figure 3A–B*). Across the genome, this resulted in 82 different groupings of embryos according to local genotype. Embryos that had evidence of a recombination event within the informative region were assigned to a genotype group according to the largest contiguous section of the region.

Differential gene expression analysis on each different embryo grouping revealed DE genes located in or close to the region of the genome that was used to define the embryo groups (*Figure 3* and *Figure 3—figure supplement 1*, *Supplementary file 4*). The $\log_2$(fold changes) of affected genes varied widely but had an absolute mean of 0.5 for the homozygous versus homozygous comparison (*Figure 3E*). This demonstrates that genes can show ASE in a wild-type context (*Figure 3C–E*).

Through these analyses, it was also possible to see the consequences of meiotic recombination in individual embryos (*Figure 3B–C*). For example, two samples (C7 and E5) showed recombination in the 31–37 Mbp region of chromosome 5 (red ovals in *Figure 3B*). The genotypes near the *myhc4* gene were the opposite of that called for the whole region and this is evident in the count plot – C7 has expression comparable with the Tu/Tu haplotype, despite being assigned AB/Tu, and E5 has expression similar to the AB/Tu samples despite being assigned Tu/Tu based on the entire 31–37 Mbp region (*Figure 3C*).

## ASE can alter interpretation of experiments

To assess what impact ASE might have on the interpretation of RNA-seq experiments, we looked at the effect on Gene Ontology (GO) enrichments if DE genes on the same chromosome as the mutation were removed from the DE gene list. To do this, we ran GO enrichment on two different gene lists for each experiment. The first list comprised all the DE genes and the second excluded genes on the same chromosome as the mutation. The gene harbouring the mutation was not removed if it was DE. It is important to note that removing all the genes on the same chromosome potentially removes genes that are misregulated due to the mutation as well as those caused by mutation-independent ASE; for almost all experiments it is not possible to distinguish between the two without further experimental analyses (see next section). The enrichment of GO terms from the two lists was compared using the Jaccard similarity coefficient (*Jaccard, 1912*).

These analyses showed that ASE could affect enriched GO terms, but that this effect was very variable (*Figure 4A*). This is not unexpected and will depend on how many of the DE genes are on the same chromosome as the mutation and whether the genes on the same chromosome contribute to any of the enriched GO terms using the full list. Experiments where there wasn't an enrichment of DE genes on the mutant chromosome generally did not show as large an effect, which again makes sense as the DE genes linked to the mutation were a smaller fraction of the gene list.

Overall, experiments with fewer DE genes showed larger effects. However, there were experiments with hundreds to thousands of DE genes where only 50% of GO terms were shared between both sets. For example, in a *sox10 t3/baz1* intercross at 36 hpf, 302 genes were DE, 32 of which were on chromosome 3 (the same chromosome as *sox10*). GO term enrichment using the full list of genes produced 92 enriched GO terms, only 49 of which were also enriched if the genes on chromosome 3 had been removed from the list (*Figure 4B–C*). In addition, 28 extra GO terms were enriched using the shorter gene list.

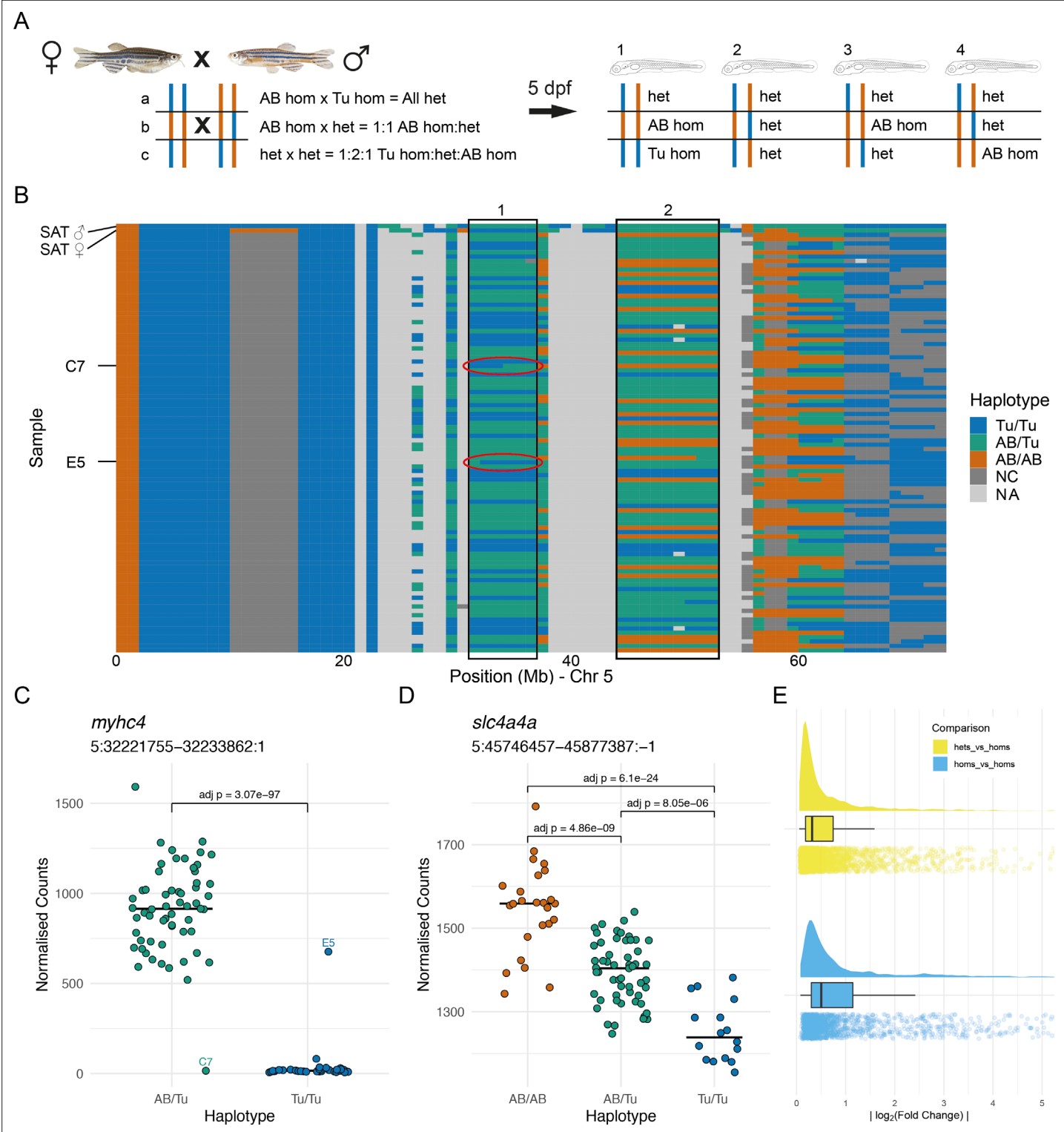

**Figure 3.** Allele-specific expression is common in wild-type embryos. (**A**) Experimental design. Two wild-type SAT fish were incrossed and 96 embryos were collected for RNA-sequencing (RNA-seq) at 5 days post-fertilisation (dpf). Depending on the haplotypes of the parents, different combinations of genotype are possible in specific regions in the offspring. (**B**) The haplotypes of the collected embryos were determined in 1 Mbp bins using the RNA-seq reads and the embryos were grouped according to the haplotypes in specific regions. Chromosome 5 is shown with chromosomal position along the x-axis and samples on the y-axis. 1 Mbp bins are coloured according to the haplotype in that region. Blue = homozygous Tübingen (Tu/Tu), green = heterozygous AB/Tübingen (AB/Tu), orange = homozygous AB (AB/AB), dark grey = not consistent with parental haplotypes (NC), light grey

*Figure 3 continued on next page*

*Figure 3 continued*

= no haplotype call (NA), due to, for example, low coverage. Examples of regions used to group the embryos are boxed. Red ovals indicate regions containing recombination breakpoints in the samples labelled in (**C**). (**C–D**) Examples of differentially expressed genes from two different groupings. (**C**) Counts for the *myhc4* gene, grouped according to the haplotypes in the region 5:31–37 Mbp (region 1 in B). The Tübingen allele is expressed at very low levels, with much higher expression in the heterozygotes. There are two examples of embryos with recombinations within the region. Compare to red ovals in the haplotype plot in (**B**). (**D**) Example of a differentially expressed gene (*slc4a4a*) in a region where all three genotypes are present (5:44–53 Mbp, region 2 in B). As in (**C**), the Tübingen allele has lower expression, with the heterozygotes showing intermediate levels. (**E**) Distribution of absolute log$_2$(fold change) values found between wild-type alleles. Differences when comparing homozygous embryos (blue) are generally larger than when comparing heterozygotes to homozygotes (yellow).

The online version of this article includes the following source data and figure supplement(s) for figure 3:

**Source data 1.** Chr5 haplotype data in the wild-type SAT cross.

**Source data 2.** Counts for myhc4.

**Source data 3.** Counts for slc4a4a.

**Source data 4.** Log$_2$ fold Change data.

**Figure supplement 1.** Allele-specific expression is linked to the region used to define the sample groupings.

## Distinguishing response genes from ASE

Having established that ASE is widespread and can significantly alter the transcriptional profiles of mutant zebrafish, we wondered whether there is a way to distinguish potential 'true' response genes located on the same chromosome as the mutation, that is, those that change expression due to the altered function of the mutated gene, from those DE genes that arise through ASE. We went back to the expression data from the compound heterozygous *sox10$^{t3}$;sox10$^{baz1}$* cross and found that the genes that were DE between *sox10$^{t3/baz1}$* individuals and their siblings and located on chromosome 3 fell into different groups with respect to their expression levels across the four different genotypes (*Figure 5*). Ten genes showed expression patterns as shown in *Figure 2C*, where increased expression was linked to the presence of a specific allele (*Figure 5A and C*). Only one gene (ENSDARG00000110416) located on another chromosome, encoding an miRNA, showed a similar pattern (*Figure 5—figure supplement 1*). By contrast, the other 15 DE genes (excluding *sox10* itself) on chromosome 3 showed genotype-dependent transcript levels that were consistent with (though do not prove) a response to loss of *sox10* function, that is, the wild types and the compound heterozygous individuals had opposing expression levels whereas both heterozygous genotypes had intermediate levels or the same as wild types (*Figure 5B and C*). *Sox10* is a key regulator of neural crest development, so we looked for published spatial expression data at 24 hpf on ZFIN (zfin.org). Of the genes we speculated to be downstream of *sox10*, all four with data on ZFIN are expressed in neural (*kctd13* and *cygb1*) and neural crest (*syngr1a* and *vasnb*) derivatives, whereas the three ASE candidates with available data are not spatially restricted (*traf7*, *polr3h,* and *polr2f*). Consequently, for genes showing single allele-linked expression patterns, it is likely that ASE is the primary driver of their differential expression and that they are probably red herrings.

## Discussion

Transcriptional profiling is a powerful and popular technique to investigate the gene expression changes resulting from organismal insults such as drug treatments, infections, or altered gene function. To gain mechanistic insight into gene regulatory events affected by a particular mutation, it is paramount to distinguish specific responses due to altered function of the mutated genes from other causes that change transcript abundance, such as developmental delay or technical artefacts such as batch effects. In this work, we describe a previously under-appreciated effect of ASE on the transcriptomes of zebrafish mutants. In 51 out of 124 transcriptional profiling experiments comparing zebrafish mutants and siblings at different stages of development, we found a statistically significant enrichment of DE genes on the same chromosome as the mutated gene. In a previous study using RNA-seq to map ENU mutations (*Miller et al., 2013*), it was noted that very few genes were detected as being DE in regions linked to the mutation. This difference is likely the result of methodological differences between the two studies, the most significant of which is the sample size. Miller et al. used one mutant and one wild-type sample, whereas our study has a median sample size of 10 per condition.

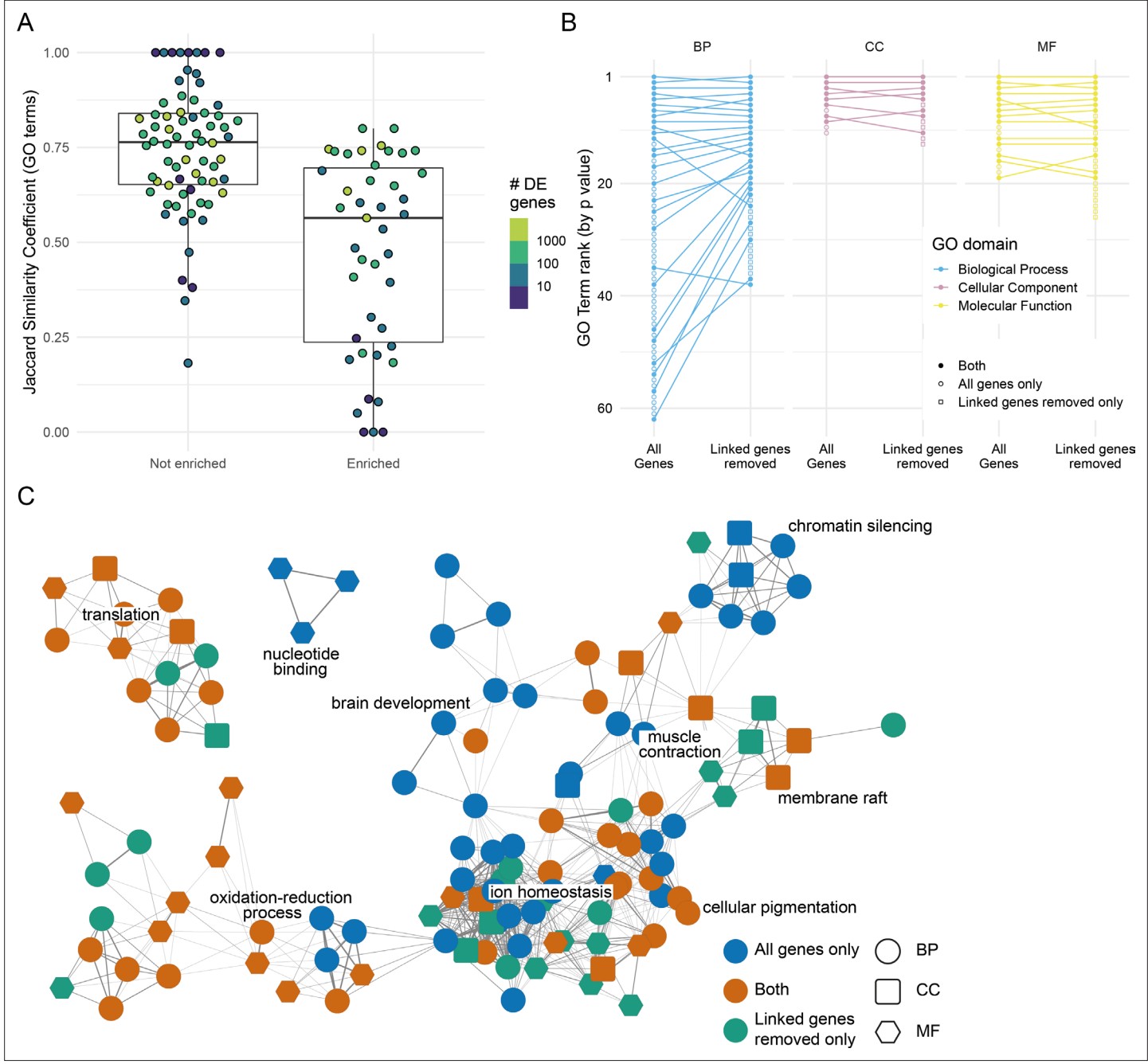

**Figure 4.** Effect of removing differentially expressed (DE) genes linked to the mutation under investigation. (**A**) Distribution of the overlap between the Gene Ontology (GO) terms enriched when DE genes linked to the mutation are removed. GO term enrichment was done on both the DE gene list and the list with the genes on the same chromosome as the mutation removed (excluding the mutated gene itself). The lists of enriched GO terms were then compared and the Jaccard similarity coefficient (number of GO terms enriched in both sets/total number of enriched GO terms) calculated. Each point represents one experiment. Experiments are split according to whether the chromosome with the mutated gene has an enrichment of DE genes or not. Points are coloured by the number of DE genes identified in the experiment ($\log_{10}$ scale). (**B**) Plot showing the changes in GO term enrichment for a single experiment (*sox10^{t3/baz1}* incross at 36 hr post-fertilisation). Each point is an enriched GO term ranked by p-value (highest ranked terms at the top) and the lines connect the same terms if they are enriched using both gene lists (all genes or genes linked to the mutation removed). Unconnected points are terms that are only enriched for either the 'all genes' list (open circles) or for the 'linked genes removed list' (open squares). (**C**) Network diagram representation of the same GO enrichments as in (**B**). Each node represents a GO term, and the nodes are connected by an edge if the genes annotated to the term overlap sufficiently (Cohen's kappa > 0.4). GO term nodes are coloured by whether they are enriched in both lists (orange) or just one (blue = all genes only, green = linked genes removed only). The shape of the nodes represents the GO term domain of the term (circle = biological process, square = cellular component, hexagon = molecular function).

*Figure 4 continued on next page*

*Figure 4 continued*

The online version of this article includes the following source data for figure 4:

**Source data 1.** Gene Ontology (GO) enrichments overlaps.

**Source data 2.** Gene Ontology (GO) enrichments for *sox10*$^{t3/baz1}$ incross at 36 hr post-fertilisation (hpf).

The physical arrangement of genes in an organism's genome is not random. Co-expression of functionally related genes using shared regulatory elements and/or transcription factors provides an evolutionary pressure to keep these genes clustered in physical proximity within a chromosome (*Thévenin et al., 2014*). Consequently, it is possible that a mutation affecting one gene could alter expression levels of nearby genes if they form a functionally related cluster. However, the neighbouring DE genes in the tested mutant lines showed no obvious functional connections. Of note, 7/16 chromosomal enrichments that were not linked to the mutated genes affected a chromosome 9 cluster of crystallin genes that are expressed in the eye. Instead we found that the enrichments were driven by ASE, which has been widely demonstrated across different tissues and organisms (*Battle et al., 2017*; *Huang et al., 2015*; *Kim-Hellmuth et al., 2020*) and can play a role in developmental and disease processes (*Libioulle et al., 2007*; *Moffatt et al., 2007*; *Nicolae et al., 2010*).

ASE is often tissue-dependent and the average log$_2$(fold change) between alleles in human ASE is about 0.6 as measured in different tissues (*Battle et al., 2017*). Here, we have observed ASE at similar magnitudes even when averaged across all tissues through whole embryo RNA-seq. This suggests that the expression differences between alleles would be even larger when looking at individual tissues.

Zebrafish wild-type 'strains' are not strains in the same sense as the well-characterised inbred lines in mouse or medaka, for example. Zebrafish are highly polymorphic, such that ASE is evident even in lines that were not outcrossed to another genetic background before the experiment. Consequently, ASE could potentially impact the penetrance or expressivity of phenotypes caused by the same mutation in different genetic backgrounds (*Sanders and Whitlock, 2003*; *Sheehan-Rooney et al., 2013*; *Young et al., 2019*). Indeed, *Sheehan-Rooney et al., 2013*, showed that the expression of *ahsa1a* differed by more than threefold in two different backgrounds (WIK and EkkWill) and was responsible for a difference in severity of the phenotype caused by a mutation in *gata3*. The effect of ASE is expected to be much less pronounced in RNA-seq data from inbred mouse strains in which allelic polymorphism is much less common. Indeed, in our work on RNA-seq data from mouse knockouts (*Collins et al., 2019*), we did not observe enrichment of DE genes on the mutant chromosome. However, ASE should be considered when working with wild mouse strains, crosses between different genetic backgrounds, or indeed with any organism that isn't fully inbred.

Given that ASE can lead to differential expression between mutants and siblings, can we correct for it in transcript profiling experiments? The solution is not as simple as removing any DE genes in the same region of the chromosome as the mutation being studied. This is because the DE genes on the same chromosome as the mutation are likely to be a mix of genuine responses to the mutation and linkage of ASE unrelated to the mutation. One way to resolve this would be to use two different mutant alleles of the same gene to generate compound heterozygotes and enable tracking of parental alleles. This would allow genotyping for both alleles and the ability therefore to also identify the different wild-type chromosomes in the cross. As shown in *Figure 5*, this makes it possible to distinguish between potential genuine responses to the mutation and spurious ones. Another possibility would be to identify an informative SNP in the wild-type alleles of the mutant gene being studied to allow genotyping of both the mutation and the wild-type alleles. There are also complementary approaches to investigate gene function that avoid the confounding effects of ASE. Transgenic overexpression of the gene of interest could validate true target gene responses and should leave ASE genes unaffected. Alternatively, analysing morpholino- or CRISPR/Cas9-injected G0 embryos (*Eisen and Smith, 2008*; *Kroll et al., 2021*; *Wu et al., 2018*) should avoid the ASE effect as the injected embryos will have a mix of background alleles. Note that although using G0 CRISPR/Cas9 mutants avoids the effect of ASE, F2 fish carrying stable CRISPR/Cas9-induced mutations will again show the effects of ASE when comparing homozygous mutants to siblings.

All these methods involve extra effort and expense, as well as having their own specific caveats and drawbacks (such as off-targets effects and mosaicism), and so would need careful consideration with respect to the need to validate specific gene expression changes for the conclusions of the study. As

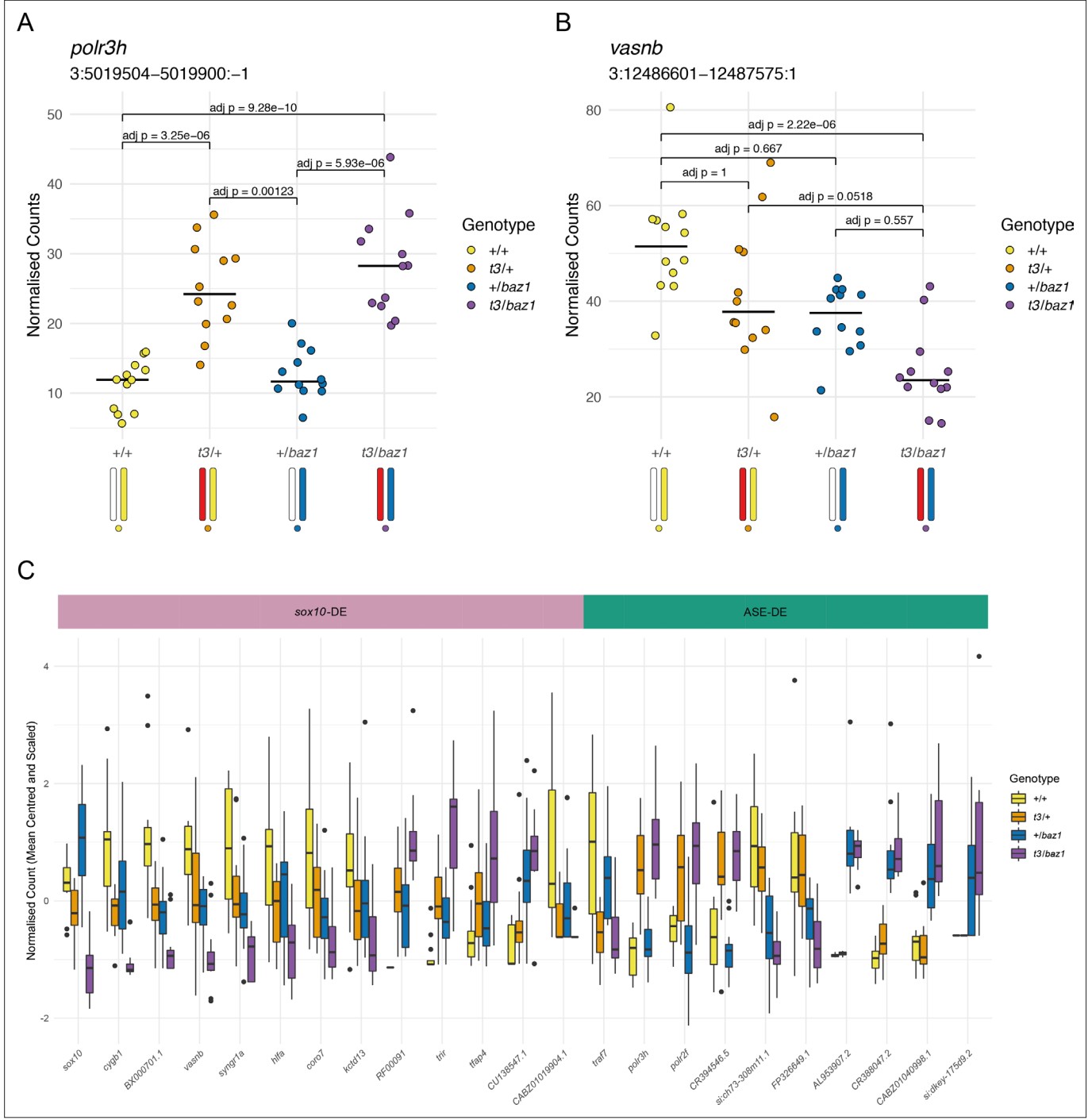

**Figure 5.** Distinguishing mutation-dependent gene expression changes from allele-specific expression (ASE). (**A**) Plot of normalised counts consistent with ASE. This shows either reduced expression from the allele on one of the wild-type chromosomes (white chromosome in the diagram under the plot) or increased expression from the allele on the *t3* chromosome (red chromosome). Yellow = wild-types (+/+), orange = *t3* heterozygotes (*t3/+*), blue = *baz1* heterozygotes (+/*baz1*), purple = compound heterozygotes (*t3/baz1*). (**B**) Normalised counts consistent with a response to the *sox10* mutations. The compound heterozygotes have reduced expression and the other two groups of heterozygotes are intermediate between the compound heterozygotes and the wild types. (**C**). Boxplots of the expression of all the differentially expressed (DE) genes on chromosome 3. These are split into two groups, those that are consistent with being downstream of *sox10* (*sox10*-DE) and those that appear to be driven by allele-specific expression unrelated to *sox10* (ASE-DE).

The online version of this article includes the following source data and figure supplement(s) for figure 5:

**Source data 1.** Counts for polr3h.

*Figure 5 continued on next page*

*Figure 5 continued*

**Source data 2.** Counts for vasnb.

**Source data 3.** Counts for the genes represented in *Figure 5C*.

**Figure supplement 1.** Allele-specific expression not linked to the homozygous region.

a first step, we recommend that, whatever analysis pipeline is used, the output of DE genes contains the locations of the genes, making it possible to easily see which genes are on the same chromosome as the mutation and therefore candidates for ASE.

# Materials and methods

**Key resources table**

| Reagent type (species) or resource | Designation | Source or reference | Identifiers | Additional information |
|---|---|---|---|---|
| Gene (zebrafish, *Danio rerio*) | mitfa | Ensembl | ENSDARG00000003732 | |
| Gene (zebrafish, *Danio rerio*) | sox10 | Ensembl | ENSDARG00000077467 | |
| Gene (zebrafish, *Danio rerio*) | si:ch73-308m11.1 | Ensembl | ENSDARG00000039752 | |
| Gene (zebrafish, *Danio rerio*) | myhc4 | Ensembl | ENSDARG00000035438 | |
| Gene (zebrafish, *Danio rerio*) | slc4a4a | Ensembl | ENSDARG00000013730 | |
| Gene (zebrafish, *Danio rerio*) | polr3h | Ensembl | ENSDARG00000102590 | |
| Gene (zebrafish, *Danio rerio*) | vasnb | Ensembl | ENSDARG00000102565 | |
| Gene (zebrafish, *Danio rerio*) | gata3 | Ensembl | ENSDARG00000016526 | |
| Gene (zebrafish, *Danio rerio*) | ahsa1a | Ensembl | ENSDARG00000028664 | |
| Gene (zebrafish, *Danio rerio*) | BX537296.1 | Ensembl | ENSDARG00000110416 | |
| Gene (zebrafish, *Danio rerio*) | cygb1 | Ensembl | ENSDARG00000099371 | |
| Gene (zebrafish, *Danio rerio*) | BX000701.1 | Ensembl | ENSDARG00000099172 | |
| Gene (zebrafish, *Danio rerio*) | syngr1a | Ensembl | ENSDARG00000002564 | |
| Gene (zebrafish, *Danio rerio*) | hlfa | Ensembl | ENSDARG00000074752 | |
| Gene (zebrafish, *Danio rerio*) | coro7 | Ensembl | ENSDARG00000089616 | |
| Gene (zebrafish, *Danio rerio*) | kctd13 | Ensembl | ENSDARG00000044769 | |
| Gene (zebrafish, *Danio rerio*) | RF00091 | Ensembl | ENSDARG00000084991 | |
| Gene (zebrafish, *Danio rerio*) | trir | Ensembl | ENSDARG00000104178 | |

*Continued on next page*

*Continued*

| Reagent type (species) or resource | Designation | Source or reference | Identifiers | Additional information |
|---|---|---|---|---|
| Gene (zebrafish, *Danio rerio*) | *tfap4* | Ensembl | ENSDARG00000103923 | |
| Gene (zebrafish, *Danio rerio*) | *CU138547.1* | Ensembl | ENSDARG00000074231 | |
| Gene (zebrafish, *Danio rerio*) | *CABZ01019904.1* | Ensembl | ENSDARG00000104193 | |
| Gene (zebrafish, *Danio rerio*) | *traf7* | Ensembl | ENSDARG00000060207 | |
| Gene (zebrafish, *Danio rerio*) | *polr2f* | Ensembl | ENSDARG00000036625 | |
| Gene (zebrafish, *Danio rerio*) | *CR394546.5* | Ensembl | ENSDARG00000112755 | |
| Gene (zebrafish, *Danio rerio*) | *FP326649.1* | Ensembl | ENSDARG00000088820 | |
| Gene (zebrafish, *Danio rerio*) | *AL953907.2* | Ensembl | ENSDARG00000113960 | |
| Gene (zebrafish, *Danio rerio*) | *CR388047.2* | Ensembl | ENSDARG00000109888 | |
| Gene (zebrafish, *Danio rerio*) | *CABZ01040998.1* | Ensembl | ENSDARG00000111638 | |
| Gene (zebrafish, *Danio rerio*) | *si:dkey-175d9.2* | Ensembl | ENSDARG00000093476 | |
| Strain, strain background (zebrafish, *Danio rerio*) | AB | ZIRC | ZDB-GENO-960809–7 | |
| Strain, strain background (zebrafish, *Danio rerio*) | Tübingen | ZIRC | ZDB-GENO-990623–3 | |
| Strain, strain background (zebrafish, *Danio rerio*) | SAT | ZIRC | ZDB-GENO-100413–1 | |
| Genetic reagent (zebrafish, *Danio rerio*) | *ENSDAR G00000089358*<sup>sa19600</sup> | PMID:23594742 | ZDB-ALT-190501–298 | |
| Genetic reagent (zebrafish, *Danio rerio*) | *bace2*<sup>hu3332</sup> | PMID:23594742 | ZDB-ALT-100723–4 | |
| Genetic reagent (zebrafish, *Danio rerio*) | *bmp7a*<sup>sa1343</sup> | PMID:23594742 | ZDB-ALT-120411–112 | |
| Genetic reagent (zebrafish, *Danio rerio*) | *cacna1c*<sup>sa6050</sup> | PMID:23594742 | ZDB-ALT-161003–17955 | |
| Genetic reagent (zebrafish, *Danio rerio*) | *capza1b*<sup>ta253a</sup> | PMID:23594742 | | Allele not cryopreserved |

*Continued*

| Reagent type (species) or resource | Designation | Source or reference | Identifiers | Additional information |
|---|---|---|---|---|
| Genetic reagent (zebrafish, *Danio rerio*) | *capzb*[hi1858bTg] | PMID:23594742 | ZDB-ALT-040907–2 | |
| Genetic reagent (zebrafish, *Danio rerio*) | *cax1*[sa10712] | PMID:23594742 | ZDB-ALT-130411–634 | |
| Genetic reagent (zebrafish, *Danio rerio*) | *cdan1*[sa5902] | PMID:23594742 | ZDB-ALT-161003–17833 | |
| Genetic reagent (zebrafish, *Danio rerio*) | *cep192*[sa875] | PMID:23594742 | ZDB-ALT-120411–491 | |
| Genetic reagent (zebrafish, *Danio rerio*) | *clp1*[sa6358] | PMID:23594742 | ZDB-ALT-161003–18184 | |
| Genetic reagent (zebrafish, *Danio rerio*) | *copb1*[sa3636] | PMID:23594742 | | Allele not cryopreserved |
| Genetic reagent (zebrafish, *Danio rerio*) | *cyfip2*[sa1556] | PMID:23594742 | ZDB-ALT-120411–193 | |
| Genetic reagent (zebrafish, *Danio rerio*) | *cylda*[sa21010] | PMID:23594742 | ZDB-ALT-161003–11078 | |
| Genetic reagent (zebrafish, *Danio rerio*) | *dag1*[hu3072] | PMID:23594742 | ZDB-ALT-070315–1 | |
| Genetic reagent (zebrafish, *Danio rerio*) | *dhx15*[sa7108] | PMID:23594742 | ZDB-ALT-161003–18741 | |
| Genetic reagent (zebrafish, *Danio rerio*) | *dmd*[ta222a] | PMID:23594742 | ZDB-ALT-980413–693 | |
| Genetic reagent (zebrafish, *Danio rerio*) | *dnmt3aa*[sa3105] | PMID:23594742 | | Allele not cryopreserved |
| Genetic reagent (zebrafish, *Danio rerio*) | *dnmt3aa*[sa617] | PMID:23594742 | ZDB-ALT-120411–432 | |
| Genetic reagent (zebrafish, *Danio rerio*) | *dnmt3ba*[sa14480] | PMID:23594742 | ZDB-ALT-130411–3189 | |
| Genetic reagent (zebrafish, *Danio rerio*) | *dnmt3bb.1*[sa15458] | PMID:23594742 | ZDB-ALT-130411–4030 | |
| Genetic reagent (zebrafish, *Danio rerio*) | *frem2a*[sa21742] | PMID:23594742 | ZDB-ALT-161003–11257 | |
| Genetic reagent (zebrafish, *Danio rerio*) | *glra1*[sa3896] | PMID:23594742 | | Allele not cryopreserved |

*Continued on next page*

*Continued*

| Reagent type (species) or resource | Designation | Source or reference | Identifiers | Additional information |
|---|---|---|---|---|
| Genetic reagent (zebrafish, *Danio rerio*) | gmds*p31erb* | PMID:23594742 | ZDB-ALT-051012–8 | |
| Genetic reagent (zebrafish, *Danio rerio*) | gpaa1*sa2042* | PMID:23594742 | ZDB-ALT-161003–10931 | |
| Genetic reagent (zebrafish, *Danio rerio*) | greb1*sa1260* | PMID:23594742 | ZDB-ALT-120411–60 | |
| Genetic reagent (zebrafish, *Danio rerio*) | grin2b (2 of 2)*sa19927* | PMID:23594742 | ZDB-ALT-190501–603 | |
| Genetic reagent (zebrafish, *Danio rerio*) | hsp90aa1.1*u45* | PMID:18256191 | ZDB-ALT-080401–1 | |
| Genetic reagent (zebrafish, *Danio rerio*) | jak2b*sa20578* | PMID:23594742 | ZDB-ALT-161003–10984 | |
| Genetic reagent (zebrafish, *Danio rerio*) | kdm2aa*sa898* | PMID:23594742 | ZDB-ALT-120727–213 | |
| Genetic reagent (zebrafish, *Danio rerio*) | kdm2aa*sa9360* | PMID:23594742 | ZDB-ALT-161003–20015 | |
| Genetic reagent (zebrafish, *Danio rerio*) | kitlga*tc244b* | PMID:23364329 | ZDB-ALT-980203–1317 | |
| Genetic reagent (zebrafish, *Danio rerio*) | lamb2*tm272a* | PMID:19736328 | ZDB-ALT-980203–1438 | |
| Genetic reagent (zebrafish, *Danio rerio*) | lamc1*sa379* | PMID:23594742 | ZDB-ALT-120411–351 | |
| Genetic reagent (zebrafish, *Danio rerio*) | las1l*sa674* | PMID:23594742 | ZDB-ALT-120727–150 | |
| Genetic reagent (zebrafish, *Danio rerio*) | ldlr*sa16375* | PMID:23594742 | ZDB-ALT-130411–4850 | |
| Genetic reagent (zebrafish, *Danio rerio*) | mapta*sa22009* | PMID:23594742 | ZDB-ALT-161003–11315 | |
| Genetic reagent (zebrafish, *Danio rerio*) | mdn1*sa1349* | PMID:23594742 | ZDB-ALT-120411–117 | |
| Genetic reagent (zebrafish, *Danio rerio*) | megf10*sa6166* | PMID:23594742 | ZDB-ALT-161003–18049 | |
| Genetic reagent (zebrafish, *Danio rerio*) | meis1*sa9839* | PMID:23594742 | ZDB-ALT-130411–5422 | |

*Continued on next page*

*Continued*

| Reagent type (species) or resource | Designation | Source or reference | Identifiers | Additional information |
|---|---|---|---|---|
| Genetic reagent (zebrafish, *Danio rerio*) | *mitfa*$^{w2}$ | PMID:10433906 | ZDB-ALT-990423–22 | |
| Genetic reagent (zebrafish, *Danio rerio*) | *neb*$^{hu2849}$ | PMID:23594742 | ZDB-ALT-070730–10 | |
| Genetic reagent (zebrafish, *Danio rerio*) | *buf*$^{ti209}$ | PMID:9007258 | ZDB-ALT-980203–1049 | |
| Genetic reagent (zebrafish, *Danio rerio*) | *nod2*$^{sa18880}$ | PMID:23594742 | ZDB-ALT-161003–10423 | |
| Genetic reagent (zebrafish, *Danio rerio*) | *nol9*$^{sa1022}$ | PMID:23594742 | ZDB-ALT-120411–10 | |
| Genetic reagent (zebrafish, *Danio rerio*) | *nol9*$^{sa1029}$ | PMID:23594742 | ZDB-ALT-160721–33 | |
| Genetic reagent (zebrafish, *Danio rerio*) | *nup88*$^{sa2206}$ | PMID:23594742 | ZDB-ALT-120727–92 | |
| Genetic reagent (zebrafish, *Danio rerio*) | *pax2a*$^{sa24936}$ | PMID:23594742 | ZDB-ALT-161003–12106 | |
| Genetic reagent (zebrafish, *Danio rerio*) | *pcna*$^{sa8962}$ | PMID:23594742 | ZDB-ALT-161003–19656 | |
| Genetic reagent (zebrafish, *Danio rerio*) | *pla2g12b*$^{sa659}$ | PMID:23594742 | ZDB-ALT-161003–18374 | |
| Genetic reagent (zebrafish, *Danio rerio*) | *pld1*$^{sa1311}$ | PMID:23594742 | ZDB-ALT-120411–91 | |
| Genetic reagent (zebrafish, *Danio rerio*) | *polr1a*$^{sa1376}$ | PMID:23594742 | ZDB-ALT-120411–135 | |
| Genetic reagent (zebrafish, *Danio rerio*) | *ptf1a*$^{sa126}$ | PMID:23594742 | ZDB-ALT-100506–17 | |
| Genetic reagent (zebrafish, *Danio rerio*) | *rpl13*$^{sa638}$ | PMID:23594742 | ZDB-ALT-161003–18201 | |
| Genetic reagent (zebrafish, *Danio rerio*) | *rps24*$^{sa2681}$ | PMID:23594742 | ZDB-ALT-161003–12995 | |
| Genetic reagent (zebrafish, *Danio rerio*) | *ryr1*$^{sa23341}$ | PMID:23594742 | ZDB-ALT-161003–11675 | |
| Genetic reagent (zebrafish, *Danio rerio*) | *ryr1*$^{sa6529}$ | PMID:23594742 | ZDB-ALT-161003–18326 | |

*Continued on next page*

*Continued*

| Reagent type (species) or resource | Designation | Source or reference | Identifiers | Additional information |
|---|---|---|---|---|
| Genetic reagent (zebrafish, *Danio rerio*) | sh3gl2*sa19508* | PMID:23594742 | ZDB-ALT-161003–10694 | |
| Genetic reagent (zebrafish, *Danio rerio*) | si:ch211-168k14.2*sa6115* | PMID:23594742 | ZDB-ALT-161003–18015 | |
| Genetic reagent (zebrafish, *Danio rerio*) | slc22a7b*sa365* | PMID:23594742 | ZDB-ALT-120411–342 | |
| Genetic reagent (zebrafish, *Danio rerio*) | slc2a11b*sa1577* | PMID:23594742 | ZDB-ALT-120411–200 | |
| Genetic reagent (zebrafish, *Danio rerio*) | smarce1*sa18758* | PMID:23594742 | Allele not cryopreserved | |
| Genetic reagent (zebrafish, *Danio rerio*) | sox10*baz1* | PMID:17065232 | ZDB-ALT-070131–1 | |
| Genetic reagent (zebrafish, *Danio rerio*) | sox10*t3* | PMID:11684650 | ZDB-ALT-980203–1827 | |
| Genetic reagent (zebrafish, *Danio rerio*) | srpk3*sa18907* | PMID:23594742 | ZDB-ALT-161003–10436 | |
| Genetic reagent (zebrafish, *Danio rerio*) | sucla2*sa20646* | PMID:23594742 | ZDB-ALT-161003–11003 | |
| Genetic reagent (zebrafish, *Danio rerio*) | tcf7l1a*m881* | PMID:11057671 | ZDB-ALT-001107–2 | |
| Genetic reagent (zebrafish, *Danio rerio*) | tfap2a*sa24445* | PMID:23594742 | ZDB-ALT-131217–17748 | |
| Genetic reagent (zebrafish, *Danio rerio*) | tfap2c*sa18857* | PMID:23594742 | Allele not cryopreserved | |
| Genetic reagent (zebrafish, *Danio rerio*) | tfip11*sa3219* | PMID:23594742 | ZDB-ALT-120727–140 | |
| Genetic reagent (zebrafish, *Danio rerio*) | tmod4*hu3530* | PMID:23594742 | ZDB-ALT-070914–1 | |
| Genetic reagent (zebrafish, *Danio rerio*) | top1l*sa2597* | PMID:23594742 | ZDB-ALT-161003–12704 | |
| Genetic reagent (zebrafish, *Danio rerio*) | ttna*sa1029* | PMID:23594742 | ZDB-ALT-160721–33 | |
| Genetic reagent (zebrafish, *Danio rerio*) | ttna*sa787* | PMID:23594742 | ZDB-ALT-120411–459 | |

*Continued*

| Reagent type (species) or resource | Designation | Source or reference | Identifiers | Additional information |
|---|---|---|---|---|
| Genetic reagent (zebrafish, *Danio rerio*) | *ttnb*sa5562 | PMID:23594742 | Allele not cryopreserved | |
| Genetic reagent (zebrafish, *Danio rerio*) | *vps16*sa7027 | PMID:23594742 | ZDB-ALT-161003–18689 | |
| Genetic reagent (zebrafish, *Danio rerio*) | *vps16*sa7028 | PMID:23594742 | ZDB-ALT-161003–18690 | |
| Genetic reagent (zebrafish, *Danio rerio*) | *vps51*p9emcf | PMID:16581006 | ZDB-ALT-060602–2 | |
| Genetic reagent (zebrafish, *Danio rerio*) | *wu:fj82b07*sa24599 | PMID:23594742 | ZDB-ALT-161003–20235 | |
| Genetic reagent (zebrafish, *Danio rerio*) | *yap1*sa25458 | PMID:23594742 | ZDB-ALT-200207–2 | |
| Genetic reagent (zebrafish, *Danio rerio*) | *zgc:171,763*sa22031 | PMID:23594742 | ZDB-ALT-161003–11320 | |
| Software, algorithm | HISAT2 | PMID:31375807 | RRID:SCR_015530 version 2.1.0 | https://github.com/DaehwanKimLab/hisat2 |
| Software, algorithm | featureCounts | PMID:24227677 | | |
| Software, algorithm | DESeq2 | PMID:25516281 | | |
| Software, algorithm | BCFTools | PMID:33590861 | RRID:SCR_002105 version 1.4 | https://samtools.github.io/bcftools/bcftools.html |
| Software, algorithm | statsmodels | http://conference.scipy.org/proceedings/scipy2010/pdfs/seabold.pdf | | https://www.statsmodels.org/stable/index.html |
| Software, algorithm | DeTCT | PMID:26238335 | | |
| Software, algorithm | BWA | https://arxiv.org/abs/1303.3997 | | |
| Software, algorithm | biobambam | https://gitlab.com/german.tischler/biobambam2 | | |
| Software, algorithm | mpileup | PMID:21903627 | | |
| Software, algorithm | QCALL | PMID:20980557 | | |
| Software, algorithm | GATK | PMID:21478889 | | |
| Software, algorithm | Tophat | PMID:23618408 | | |
| Software, algorithm | QoRTs | PMID:26187896 | | |

## RNA-seq and LD mapping

Eight independent mutant fish lines under study by groups at UCL (zebrafishucl.org) were analysed by RNA-seq in order to simultaneously gain gene expression data and to measure alleles across the genome in order to help map the causative mutation. Seven of these lines were the product of ENU random mutagenesis, one was created by a random viral insertion, and one by a targeted CRISPR insertion. An additional sample was taken from the literature (*Armant et al., 2016*) at random by searching Pubmed for papers where RNA-seq data had been uploaded to the European Nucleotide Archive.

In preparation for RNA-seq, embryos or larvae were sorted into two groups based on their phenotypes (mutant and sibling), each comprising three pools of at least 15 individuals. RNA was extracted from these six samples and sequenced using the Illumina NextSeq platform (2 × 75 bp reads, approximately 75 million reads per sample). Reads were aligned to the GRCz10 genome using HISAT2 (*Kim et al., 2019*). To measure differential expression, transcripts were counted from the aligned RNA-seq reads using featureCounts (*Liao et al., 2014*) and compared using DESeq2 (*Love et al., 2014*). A gene was considered DE if the adjusted p-value from DESeq2 was below 0.05.

To perform LD mapping, the three samples in each group were analysed as a single pooled sample for single nucleotide polymorphisms (SNPs) by BCFtools (*Li, 2011*), calculating the allele ratio at each SNP location. SNPs which appeared in only one of the two genotype pools were filtered out, as were those with a quality score below 100. The absolute difference between a given SNP's mutant and sibling allele ratio indicates the degree of segregation of that allele (*Mackay and Schulte-Merker, 2014*). These values can be smoothed using LOESS, producing maps of the genome showing regions of high LD (*Minevich et al., 2012*). The physical location of each gene's start codon in the GRCz10 genome assembly was downloaded from Ensembl BioMart and appended to the DESeq2 table. The LD value was estimated at each gene's position based on interpolation of the LOESS-smoothed SNP data. Finally, a logistic regression model was used to test the effect of LD on a gene's probability of being DE. This was performed using the Logit function of the Python module statsmodels.

## DeTCT sequencing

DeTCT libraries were generated, sequenced, and analysed as described previously (*Collins et al., 2015*). The resulting genomic regions and putative 3′ ends were filtered using DeTCT's filter_output (v0.2.0)script (https://github.com/iansealy/DETCT/blob/master/script/filter_output.pl, *Sealy, 2020*) in its `--strict` mode. `--strict` mode removes 3′ ends in coding sequence, transposons, if nearby sequence is enriched for As or if not near a primary hexamer. Regions not associated with 3′ ends are also removed. Differential expression analysis was done using DeTCT's run_pipeline (v0.2.0)script (https://github.com/iansealy/DETCT/blob/master/script/run_pipeline.pl) using DESeq2 (*Love et al., 2014*) with an adjusted p-value cut-off of 0.05. Sequence data were deposited in the European Nucleotide Archive (ENA) under accessions ERP001656, ERP004581, ERP006132, ERP003802, ERP004579, ERP005517, ERP008771, ERP005564, ERP009868, ERP006133, ERP009078, and ERP013835. Details on the experiments are in *Supplementary file 5*.

## DNA sequencing

Double haploid AB and Tübingen fish were produced and sequenced as described in *Howe et al., 2013*. Whole genome sequencing data (SRA Study: ERP000232) was downloaded from the European Nucleotide Archive. Exome sequencing on parents for the wild-type SAT cross was done as described (*Kettleborough et al., 2013*). Reads were mapped to the GRCz11 genome assembly using BWA (*Li and Durbin, 2010*, v0.5.10) and duplicates were marked with biobambam (*Tischler and Leonard, 2014*). SNPs were called using a modified version of the 1000 Genomes Project variant calling pipeline (*Abecasis et al., 2010*). Initial calls were done by SAMtools mpileup (*Li, 2011*), QCALL (*Le and Durbin, 2011*), and the GATK Unified Genotyper (*DePristo et al., 2011*). SNPs not called by all three callers were removed from the analysis, along with any SNP that did not pass a caller's standard filters. Additionally, SNPs were removed where the genotype quality was lower than 100 for GATK and lower than 50 for QCALL and SAMtools mpileup and where the mean read depth per sample was less than 10. These SNP calls were then filtered for positions that are informative of the parental background in the SAT cross, that is, ones that are homozygous reference in one double haploid fish and homozygous alternate in the other.

## RNA-seq of wild-type SAT embryos

RNA was extracted from 5 dpf larvae as described previously (*Wali et al., 2022*). Briefly, RNA was extracted from individual embryos by mechanical lysis in RLT buffer (Qiagen) containing 1 μl of 14.3 M β-mercaptoethanol (Sigma). The lysate was combined with 1.8 volumes of Agencourt RNAClean XP (Beckman Coulter) beads and allowed to bind for 10 min. The plate was applied to a plate magnet (Invitrogen) until the solution cleared and the supernatant was removed without disturbing the beads. This was followed by washing the beads three times with 70% ethanol. After

the last wash, the pellet was allowed to air-dry for 10 min and then resuspended in 50 µl of RNAse-free water. RNA was eluted from the beads by applying the plate to the magnetic rack. Samples were DNase-I treated to remove genomic DNA. RNA was quantified using Quant-IT RNA assay (Invitrogen). Stranded RNA-seq libraries were constructed using the Illumina TruSeq Stranded RNA protocol after treatment with Ribozero. Libraries were pooled and sequenced on six Illumina HiSeq 2500 lanes in 75 bp paired-end mode. Sequence data were deposited in ENA under accession ERP011556. Reads for each sample were aggregated across lanes (median reads per embryo = 18.1 M) and mapped to the GRCz11 zebrafish genome assembly using TopHat (*Kim et al., 2013*, v2.0.13, options: `--library-type` fr-firststrand). The data were assessed for technical quality (GC-content, insert size, proper pairs, etc.) using QoRTs (*Hartley and Mullikin, 2015*). Counts for genes were produced using htseq-count (*Anders et al., 2015*, v0.6.0 options: `--stranded` = reverse) with the Ensembl v97 annotation as a reference. Differential expression analysis was done in R (*R Development Core Team, 2019*) with DESeq2 (*Love et al., 2014*) using a cut-off for adjusted p-values of 0.05.

The samples were genotyped at the positions that were determined to be informative using the double haploid sequence using GATK's SplitNCigarReads tool followed by the HaplotypeCaller (*Poplin et al., 2017*) on the RNA-seq data. The genotype calls were converted to their strain of origin (either DHAB or DHTu) and haplotypes were called by taking the most frequent genotype call in 1 Mbp windows. Any haplotypes that were not consistent with the parental haplotypes were removed.

## Acknowledgements

We thank G Gestri, L Tucker, R Martinho, A Faro, G Powell, M Khosravi, I Barlow, J Rihel and T Hawkins for providing RNA-seq datasets from mutant lines, Alex Nechiporuk for providing mutant lines, technical staff in our fish facilities for animal care, and Ian Sealy and Munise Merteroglu for critical reading of the manuscript. This study was supported by MRC Programme Grant support to Gaia Gestri and SW (MR/L003775/1 and MR/T020164/1), and a Wellcome Trust Investigator Award to SW (095722/Z/11/Z). EBN and RJW were supported by core funding to the Sanger Institute by the Wellcome Trust (206194).

## Additional information

### Funding

| Funder | Grant reference number | Author |
| --- | --- | --- |
| Medical Research Council | MR/L003775/1 | Stephen W Wilson |
| Medical Research Council | MR/T020164/1 | Stephen W Wilson |
| Wellcome Trust | 095722/Z/11/Z | Stephen W Wilson |
| Wellcome Trust | 206194 | Richard J White Elisabeth M Busch-Nentwich |

The funders had no role in study design, data collection and interpretation, or the decision to submit the work for publication.

### Author contributions

Richard J White, Eirinn Mackay, Formal analysis, Software, Visualization, Writing - original draft, Writing - review and editing; Stephen W Wilson, Elisabeth M Busch-Nentwich, Conceptualization, Funding acquisition, Supervision, Writing - original draft, Writing - review and editing

### Author ORCIDs

Richard J White http://orcid.org/0000-0003-1842-412X
Eirinn Mackay http://orcid.org/0000-0003-1436-2259
Stephen W Wilson http://orcid.org/0000-0002-8557-5940
Elisabeth M Busch-Nentwich http://orcid.org/0000-0001-6450-744X

**Decision letter and Author response**

Decision letter https://doi.org/10.7554/eLife.72825.sa1

Author response https://doi.org/10.7554/eLife.72825.sa2

---

# Additional files

## Supplementary files

• Supplementary file 1. Linkage disequilibrium (LD) mapping plots for the remaining eight lines.

• Supplementary file 2. Chromosome enrichment results. Binomial test results for each chromosome in each experiment.

• Supplementary file 3. Differentially expressed genes on chromosome 9 from seven experiments that show an enrichment of differentially expressed genes on chromosome 9.

• Supplementary file 4. Summary of differential expression numbers by genomic region in wild-type SAT RNA-sequencing (RNA-seq).

• Supplementary file 5. Summary information of DeTCT experiments.

• Transparent reporting form

## Data availability

Sequencing data have been deposited in ENA under the accessions shown in the Materials and Methods. Differentially expressed gene lists for all the experiments are available at https://doi.org/10.6084/m9.figshare.15082239.

The following dataset was generated:

| Author(s) | Year | Dataset title | Dataset URL | Database and Identifier |
|---|---|---|---|---|
| White et al | 2016 | Transcriptome_profiling_ of_zebrafish_embryos_ from_the_SAT__Sanger_ AB_T_bingen__strain | https://www.ebi.ac. uk/ena/browser/view/ PRJEB10320?show= reads | ENA, ERP011556 |

The following previously published datasets were used:

| Author(s) | Year | Dataset title | Dataset URL | Database and Identifier |
|---|---|---|---|---|
| Dooley et al | 2019 | Transcriptome_profiling_ of_zebrafish_neural_crest_ mutants | https://www.ebi.ac. uk/ena/browser/view/ PRJEB4509?show= reads | ENA, ERP003802 |
| Dooley et al | 2019 | Transcriptome_profiling_ of_zebrafish_neural_crest_ mutants | https://www.ebi.ac. uk/ena/browser/view/ PRJEB5202?show= reads | ENA, ERP004579 |
| Dooley et al | 2019 | Transcriptome_profiling_ of_zebrafish_neural_crest_ mutants | https://www.ebi.ac. uk/ena/browser/view/ PRJEB6055?show= reads | ENA, ERP005517 |
| Dooley et al | 2019 | Transcriptome_profiling_ of_zebrafish_neural_crest_ mutants | https://www.ebi.ac. uk/ena/browser/view/ PRJEB7799?show= reads | ENA, ERP008771 |
| Kettleborough et al | 2013 | Transcriptome_profiling_ of_mutants_from_the_ zebrafish_genome_project | https://www.ebi.ac. uk/ena/browser/view/ PRJEB3181?show= reads | ENA, ERP001656 |

*Continued on next page*

*Continued*

| Author(s) | Year | Dataset title | Dataset URL | Database and Identifier |
|---|---|---|---|---|
| Kettleborough et al | 2013 | Transcriptome_profiling_ of_mutants_from_the_ zebrafish_genome_project | https://www.ebi.ac. uk/ena/browser/view/ PRJEB5204?show= reads | ENA, ERP004581 |
| Kettleborough et al | 2013 | Transcriptome_profiling_ of_embryos_collected_ for_one_or_more_alleles_ identified_by_the_ zebrafish_mut | https://www.ebi.ac. uk/ena/browser/view/ PRJEB6584?show= reads | ENA, ERP006132 |
| Kettleborough et al | 2014 | Transcriptome_profiling_ of_zebrafish_muscle_ mutants | https://www.ebi.ac. uk/ena/browser/view/ PRJEB6097?show= reads | ENA, ERP005564 |
| Kettleborough et al | 2015 | Transcriptome_profiling_ of_zebrafish_muscle_ mutants | https://www.ebi.ac. uk/ena/browser/view/ PRJEB8827?show= reads | ENA, ERP009868 |
| Kettleborough et al | 2014 | Transcriptome_profiling_ of_embryos_genotyped_ for_one_or_more_alleles_ in_genes_involved_in_ DNA_methyl | https://www.ebi.ac. uk/ena/browser/view/ PRJEB6585?show= reads | ENA, ERP006133 |
| Kettleborough et al | 2015 | Transcriptome_profiling_ of_zebrafish_metabolic_ mutants | https://www.ebi.ac. uk/ena/browser/view/ PRJEB8043?show= reads | ENA, ERP009078 |
| Kettleborough et al | 2016 | Transcriptome_profiling_ of_zebrafish_hesx1_ knockout_embryos | https://www.ebi.ac. uk/ena/browser/view/ PRJEB12364?show= reads | ENA, ERP013835 |
| Howe et al | 2010 | The Sequence of the Two Most Common Zebrafish Laboratory Strains: AB and Tuebingen | https://www.ebi.ac. uk/ena/browser/view/ PRJEB2177?show= reads | ENA, ERP000232 |

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
