## [Editor Report]

Zebrafish strains are typically considerably polymorphic. White and colleagues tested the hypothesis that genes in linkage with a mutant allele might show allele-specific expression differences and thus potentially confound the interpretation of mutant effects. Using a variety of mutant and wild-type alleles with sophisticated analysis of RNA-seq data in zebrafish embryos they demonstrate over-representation of expression changes of genes in linkage with the mutant allele on the same chromosome. The authors provide Gene Ontology analyses to demonstrate how the allele-specific expression differences may impact on the interpretation of differential gene expression caused by a mutation. The data are extensive, carefully analysed and of sufficient depth and quality to support their main claim of frequent occurrence of allele-specific gene expression in outcross experiments. The findings of this study will be of interest to geneticists working with zebrafish strains or with strains of other polymorphic species.

---

## [Decision Letter]

**Decision letter after peer review:**

Thank you for submitting your article "A red herring in zebrafish genetics: allele-specific gene expression can underlie altered transcript abundance in zebrafish mutants" for consideration by *eLife*. Your article has been reviewed by 3 peer reviewers, including Ferenc Muller as Reviewing Editor and Reviewer #3, and the evaluation has been overseen by Didier Stainier as the Senior Editor. The following individual involved in review of your submission has agreed to reveal their identity: Shawn M Burgess (Reviewer #1).

Essential revisions:

1) The reviewers agreed that the authors need to strengthen the claim of the importance of misjudgement of differentially expressed genes in interpretation of RNA-seq data in order to draw conclusions on mutant gene function. The authors need to demonstrate using one of their analysed mutants (e.g. sox10 or lama1) how downstream conclusions on the function of the mutated gene drawn from the DE gene analysis may be affected if the allele-specific expression of genes in LD were not removed from the DE gene list. Such analysis does not require any wet lab work and could involve one of any of the following examples of non-comprehensive options of meta-analyses such as assessment of anatomy term enrichment, various forms of GO analysis, genetic or biochemical pathway analysis etc. of gene lists with and without what they call 'the red herrings'.

2) The authors need to discuss the publication by Miller et al., (Genome Research 2013) in the context of the distinct findings of the current manuscript.

*Reviewer #1 (Recommendations for the authors):*

In the discussion, I think it would be reasonable to make some more and generally declarative statements about what researchers should to either avoid these issues, control for them, or correct them.

One such argument might be to recommend an "orthogonal" KO approach such as morpholino or CRISPR mutagenesis directly in the injected embryos, and compare transcriptional profiling between approaches. Perhaps also discuss that LD blocks showing changes could still be a result of the mutation directly if it impacts gene expression in the entire chromosomal region as a cis regulator.

These are small concerns and do not impact my general enthusiasm for the manuscript.

*Reviewer #2 (Recommendations for the authors):*

It would strengthen the paper to analyze the impact of the possibly misinterpreted differentially expressed genes near the mutant locus on the conclusions drawn from the RNA seq datasets.

*Reviewer #3 (Recommendations for the authors):*

(1.) Some evaluation of the impact of differential gene expression (DE) due to allelic variation arising independently from the function of the mutated gene would be helpful. To do so the following questions may be possible to address:

a. Is it possible to demonstrate the distinct biological relevance of the allelic variation-dependent DE genes from that of gene expression changes attributed to mutation effect (e.g. in the case of the sox10 example)?

b. To what degree do the DE genes arising from LD differ in their expression pattern (e.g. ZFIN anatomical terms), GO enrichment, or association with gene regulatory network / biochemical pathway from that of the mutated gene or that of DE genes resulting from the mutation?

(2.) While it cannot be expected that the study directly addresses the following question, nevertheless, it might be worth discussing whether allelic variation-dependent DE can function as enhancer or suppressor of a mutation and can they potentially explain phenotypic difference upon loss of function mutation of genes generated in different strains?

---

## [Author Response]

Essential revisions:1) The reviewers agreed that the authors need to strengthen the claim of the importance of misjudgement of differentially expressed genes in interpretation of RNA-seq data in order to draw conclusions on mutant gene function. The authors need to demonstrate using one of their analysed mutants (e.g. sox10 or lama1) how downstream conclusions on the function of the mutated gene drawn from the DE gene analysis may be affected if the allele-specific expression of genes in LD were not removed from the DE gene list. Such analysis does not require any wet lab work and could involve one of any of the following examples of non-comprehensive options of meta-analyses such as assessment of anatomy term enrichment, various forms of GO analysis, genetic or biochemical pathway analysis etc. of gene lists with and without what they call 'the red herrings'.

Thank you for this excellent suggestion. For a more comprehensive view we have performed an analysis across experiments. This new section (Figure 4, Lines 188-216) looks at the effect on GO enrichments when genes on the same chromosome as the mutation are removed compared with the full gene list. We find that in the same way as the effect of ASE is variable from experiment to experiment, so is the result of including or excluding the potential red herring genes. As can be expected, longer lists of differentially expressed (DE) genes are more resilient to the effect, although substantial changes to GO enrichments can also occur with long DE gene lists.

2) The authors need to discuss the publication by Miller et al., (Genome Research 2013) in the context of the distinct findings of the current manuscript.

We have added a section to the Discussion (Lines 253-257) and also edited the Discussion slightly for brevity and clarity.

Reviewer #1 (Recommendations for the authors):In the discussion, I think it would be reasonable to make some more and generally declarative statements about what researchers should to either avoid these issues, control for them, or correct them.One such argument might be to recommend an "orthogonal" KO approach such as morpholino or CRISPR mutagenesis directly in the injected embryos, and compare transcriptional profiling between approaches. Perhaps also discuss that LD blocks showing changes could still be a result of the mutation directly if it impacts gene expression in the entire chromosomal region as a cis regulator.

This is a great suggestion. Using either of these techniques would mean that the ASE is effectively averaged out by the mixed genetic background of the injected embryos. We have added this to the Discussion (Lines 303-310) and made it clear that incrossing stable CRISPR lines (rather than studying injected G0 embryos) would still be subject to the effect of ASE.

These are small concerns and do not impact my general enthusiasm for the manuscript.Reviewer #2 (Recommendations for the authors):It would strengthen the paper to analyze the impact of the possibly misinterpreted differentially expressed genes near the mutant locus on the conclusions drawn from the RNA seq datasets.

We have added a section to the results as detailed above (Lines 188-216).

Reviewer #3 (Recommendations for the authors):(1.) Some evaluation of the impact of differential gene expression (DE) due to allelic variation arising independently from the function of the mutated gene would be helpful. To do so the following questions may be possible to address:a. Is it possible to demonstrate the distinct biological relevance of the allelic variation-dependent DE genes from that of gene expression changes attributed to mutation effect (e.g. in the case of the sox10 example)?b. To what degree do the DE genes arising from LD differ in their expression pattern (e.g. ZFIN anatomical terms), GO enrichment, or association with gene regulatory network / biochemical pathway from that of the mutated gene or that of DE genes resulting from the mutation?

As well as looking at GO enrichments as mentioned above, we have looked at the expression patterns of the differentially expressed genes on chromosome 3 in the *sox10* experiment where we have a good understanding of which genes are downstream of *sox10* and which aren’t. For those where expression data exists on ZFIN, the ASE genes are not spatially restricted at 24 hpf whereas those downstream of *sox10* are expressed in the nervous system and neural crest. We have included this information at the end of the Results section (Lines 234-239).

(2.) While it cannot be expected that the study directly addresses the following question, nevertheless, it might be worth discussing whether allelic variation-dependent DE can function as enhancer or suppressor of a mutation and can they potentially explain phenotypic difference upon loss of function mutation of genes generated in different strains?

Different expression levels of modifier genes could well explain differences in phenotypic severity of mutants. Indeed, we cite an example of a mutation (*gata3^b1075^*) with different severity in two different backgrounds, where the severity was associated with a difference in expression of a gene (*ahsa1a*) which acts as a suppressor of the phenotype. We have included this in the Discussion (Lines 279-284).